# Many-body simulation of two-dimensional electronic spectroscopy of excitons and trions in monolayer transition metal dichalcogenides

Roel Tempelaar [1] & Timothy C. Berkelbach [1,2]

Indications of coherently interacting excitons and trions in doped transition metal dichalcogenides have been measured as quantum beats in two-dimensional electronic spectroscopy, but the microscopic principles underlying the optical signals of exciton-trion coherence remain to be clarified. Here we present calculations of two-dimensional spectra of such monolayers based on a microscopic many-body formalism. We use a parameterized band structure and a static model dielectric function, although a full ab initio implementation of our formalism is possible in principle. Our simulated spectra are in excellent agreement with experiments, including the quantum beats, while revealing the interplay between excitons and trions in molybdenum- and tungsten-based transition metal dichalcogenides. Quantum beats are confirmed to unambiguously reflect the exciton-trion coherence time in molybdenum compounds, but are shown to provide a lower bound to the coherence time for tungsten analogues due to a destructive interference from coexisting singlet and triplet trions.

[1] Department of Chemistry, Columbia University, New York, NY 10027, USA. [2] Center for Computational Quantum Physics, Flatiron Institute, New York, NY 10010, USA. Correspondence and requests for materials should be addressed to R.T. (email: r.tempelaar@gmail.com) or to T.C.B. (email: tim.berkelbach@gmail.com)

Atomically thin materials exhibit unique physical phenomena emerging from extreme dimensional constraints, which add to their attractiveness as functional components in ultrathin electronics and optoelectronics[1]. Of particular recent interest are monolayer transition metal dichalcogenides (TMDCs), compounds of type $MX_2$, where $M$ is a transition metal and $X$ represents a chalcogen atom. Monolayer TMDCs are direct bandgap semiconducting analogs[2,3] of graphene in which charge carriers experience a large effective mass and reduced dielectric screening, resulting in strong Coulomb interactions and large exciton binding energies[4–6]. The strong Coulomb interactions also lead to the formation of higher-order excitonic complexes such as trions[7,8], biexcitons[9], and potentially Fermi polarons at large doping[10]. Excitons are known to follow a non-hydrogenic Rydberg series[4] and form in momentum valleys centered at the (inequivalent) $K$ and $K'$ points of the hexagonal Brillouin zone with wavefunctions primarily composed of transition metal $d$ orbitals[11]. Such states exhibit robust valley and spin coherence due to the sizable spin–orbit coupling[11,12]. In addition, inversion symmetry breaking results in valley-dependent optical selection rules. In particular, circularly polarized light has been shown to allow for valley-selective excitation[11,13,14].

While the steady-state properties of TMDCs have been studied in detail by linear optical techniques, the recent application of time-resolved nonlinear spectroscopy has enabled the study of excited-state dynamics on femtosecond timescales. In particular, two-dimensional electronic spectroscopy (2DES)[15], which has found extensive use in the study of molecular assemblies[16–19], has been applied to TMDCs only quite recently[20–26]. 2DES is a four-wave mixing technique that improves over two-pulse pump-probe spectroscopy in its ability to map out the full third-order optical susceptibility of a sample by correlating excitation and detection frequencies. Through this approach, indications have been observed of coherent interaction between excitons and trions in TMDCs, in the form of spectral cross-peak beatings on the 100 fs timescale[23]. However, the microscopic mechanisms underlying such quantum beats remain to be clarified.

Despite recent progress[27–29], employing first-principles techniques capable of simulating the nonlinear optical susceptibilities of condensed-phase materials has remained challenging, in particular for the full set of four-wave mixing signals contained in 2DES. This is in stark contrast to linear spectroscopy, where time-dependent density functional theory and the Bethe–Salpeter equation both predict accurate spectra, including excitonic effects[30–33]. Simulating nonlinear spectroscopic signals of trions in atomically thin materials presents further challenges due to the larger trion Hilbert space and the dense Brillouin zone sampling required to resolve the dielectric function[34,35].

Here, we present a many-body computational framework for the simulation of 2DES and apply it to the coherent interaction of trions and excitons in monolayer TMDCs. Although the approach can be straightforwardly implemented in a fully first-principles manner, here we use a parameterization of the low-energy band structure and a model dielectric function, both of which could be obtained from a calculation using the GW approximation[36]. The present work builds on an extension of the Bethe–Salpeter equation to simulate linear spectra of three-body excitonic complexes[37], combined with a Brillouin zone truncation scheme previously applied to excitons[35]. We find our simulations to accurately reproduce experimental 2D spectra, including the quantum beats observed at the cross-peak locations, allowing us to study the underlying coherent phenomena at a microscopic level. For molybdenum-based TMDCs, quantum beats are confirmed to accurately reflect the exciton–trion coherence time. In contrast, they are shown to provide a lower bound to the coherence time for tungsten analogs due to a destructive interference from coexisting singlet and triplet trions.

## Results

**Bound excitons and trions in TMDCs.** Recent interest in TMDCs has mostly focused on compounds based on molybdenum and tungsten as transition metals, and sulfur and selenium as chalcogens. Across these compounds, the quasiparticle band structure is particularly distinct for molybdenum-based ($MoX_2$) and tungsten-based ($WX_2$) TMDCs. This is illustrated in Fig. 1, showing a schematic of the band structures near the $K$ and $K'$ points, highlighting the spin–orbit splitting of the conduction bands[12]. The splitting of the valence bands is an order of magnitude larger than that of the conduction bands, and results in two distinct absorptive transitions observable in the exciton spectrum. We restrict ourselves to the lowest-energy transition (referred to as $A$ exciton) and its associated negatively charged trion complexes to focus on the interpretation of experiments that

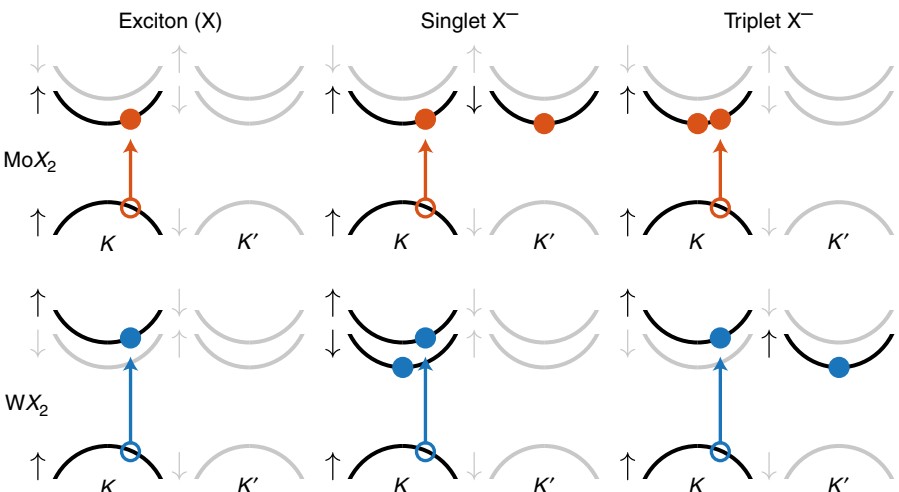

**Fig. 1** Schematic illustration of the band structure and optical transitions of monolayer $MoX_2$ and $WX_2$. Shown are the spin-dependent band structures near the $K$ and $K'$ points, derived from first-principles calculations[12], including the (helicity-selective) optical transitions and electronic configurations involved in $A$ excitons and associated negatively charged trions (the lower spin–split valence band is not shown). Singlet trions are intervalley in $MoX_2$ but intravalley in $WX_2$; the situation is reversed for triplet trions. Note that the depicted vertical transitions become mixed as a result of quasiparticle interactions

energy selectively excite this transition, although noting that a generalization to the other transition ($B$ exciton) is straightforward. We furthermore consider helicity-selective excitation in the $K$ valley; identical results would be obtained for the $K'$ valley upon flipping the spins of the involved quasiparticles. As is well known, the combination of energy and helicity selectivity to probe $A$ excitons in the $K$ valley effectively corresponds to spin-selective excitation, since the involved quasiparticles are constrained to a well-defined spin (spin-up, following the convention of Fig. 1).

Negatively charged trions, consisting of two conduction band electrons interacting with one hole in the valence band, are commonly characterized based on the spin of the two electrons, leading to singlet or triplet trions when the spins are opposite or equal, respectively. In TMDCs, the combination of valley and spin selectivity implies selection rules for trions at low temperatures (with the thermal energy smaller than the conduction band splitting). The behavior is illustrated in Fig. 1, which considers selective excitation of the $A$ transition in the $K$ valley, creating an electron–hole pair in addition to an initial one-electron state. At low temperatures, the initial electron is relaxed in the conduction band minimum of either the spin-down state in the $K'$ valley or the spin-up state in the $K$ valley. As a result, the corresponding *intervalley* trions have singlet spin, whereas *intravalley* trions have triplet spin. It is easily verified that the opposite relation holds for W$X_2$. We note that the excited electrons have identical valley and spin states only for the triplet trion in Mo$X_2$, and (repulsive) exchange interactions between conduction band electrons are therefore expected to be strongest in this case.

Since the many-body Hamiltonian conserves momentum, the trion states can be expressed as

$$|\Psi^\alpha\rangle = \sum_{c_1,c_2,v} \sum_{\mathbf{k}_1,\mathbf{k}_2} C^\alpha_{c_1,c_2,v}(\mathbf{k}_1,\mathbf{k}_2)\, a_{v,\mathbf{k}_1+\mathbf{k}_2-\mathbf{Q}}\, a^\dagger_{c_2,\mathbf{k}_2} a^\dagger_{c_1,\mathbf{k}_1}|0\rangle, \quad (1)$$

where $c_1$, $c_2$, and $v$ label the conduction and valence bands (including spin) and with $\mathbf{Q}$ as the momentum of the initial conduction band electron. In our simulation, details of which can be found in Supplementary Methods, the band structure is described by a parameterized two-band model[11,37,38], motivated by prior work showing negligible differences in optical properties compared to a more sophisticated three-band analog[11,38]. The trion states are calculated by configuration interaction using a many-body Hamiltonian containing a screened Coulomb coupling term, as done in previous extensions of the Bethe–Salpeter equation to three-particle complexes[37,39,40]. The screened Coulomb interaction is approximated as orbital-independent and isotropic, using a model dielectric function, $W(\mathbf{q}) = 2\pi e^2/q\varepsilon(q)$ with $\varepsilon(q) = 1 + 2\pi\chi_{2D}q$, where $\chi_{2D}$ is the material-dependent two-dimensional polarizability[4,38,41,42]. For two-dimensional materials, a very dense sampling of the Brillouin zone is required for convergence[35]; however, such a dense sampling makes the trion Hilbert space prohibitively large. To overcome this obstacle, we used a uniform $N \times N$ Monkhorst–Pack mesh with a cut-off radius around the $K$ and $K'$ points, denoted $k_0$. Employed previously by Qiu et al.[35] for excitons, this truncation scheme utilizes the valley confinement of low-energy excited states, and results in two convergence parameters, $N$ and $k_0$.

We first consider the exciton and trion binding energies predicted by this approach. The results are summarized in Table 1, while details are presented in Supplementary Figs. 1 and 2. The exciton binding energies are found to rapidly converge with $k_0$, with near convergence reached already for $k_0 = 0.10$ (in units of the inverse lattice constant, $2\pi/a$ (The inverse lattice constants for MoS$_2$, WS$_2$, MoSe$_2$, and WSe$_2$ are 1.97, 1.96, 1.90, and 1.90 Å$^{-1}$, respectively.)). However, convergence with $N$ is

**Table 1 Exciton and trion binding energies predicted by our model**

| | Exciton (eV) | Singlet X$^-$ (meV) | Triplet X$^-$ (meV) |
|---|---|---|---|
| MoS$_2$ | 0.53 (0.55) | 31 (34) | Unbound |
| WS$_2$ | 0.50 (0.52) | 40 (34) | 40 |
| MoSe$_2$ | 0.46 (0.51) | 27 (28) | Unbound |
| WSe$_2$ | 0.45 (0.47) | 37 (30) | 37 |

Shown are extrapolations of our calculated data to $N = \infty$. Numbers within parentheses are exact results from ref. [43]. Note that the results shown here do not include repulsive interactions between conduction and valence band electrons (see text)

very slow, with $N$ ranging from a few tens to a few hundred. Extrapolation of our results to $N = \infty$ yields exciton binding energies in reasonable agreement with those obtained in a numerically exact diffusion Monte Carlo study of the closely related real-space exciton problem[43], with 0.53 eV (vs. 0.55 eV) for MoS$_2$, 0.50 eV (0.52 eV) for WS$_2$, 0.46 eV (0.51 eV) for MoSe$_2$, and 0.45 eV (0.47 eV) for WSe$_2$. In contrast, the trion binding energies depend only weakly on $N$, suggesting a cancellation of sampling errors between the total exciton and trion energies, while a modest dependence on $k_0$ is found. For MoS$_2$ and MoSe$_2$, our model predicts singlet trion binding energies of 31 and 27 meV, respectively, whereas the triplet trion is found to be unbound as a result of the repulsive interactions between conduction band electrons. These interactions are negligible for tungsten-based TMDCs, where we find bound singlet and triplet trions with virtually identical binding energies of 40 and 37 meV for WS$_2$ and WSe$_2$, respectively. We note, however, that modest energetic splitting between these states[44–47] is in principle possible due to exchange interactions involving conduction and valence band electrons not considered in our model. The agreement with diffusion Monte Carlo results (singlet trion binding energies of 34 meV for both for MoS$_2$ and WS$_2$, 28 meV for MoSe$_2$, and 30 meV for WSe$_2$)[43] is again reasonable.

**Linear absorption and doping dependence.** Figure 2a presents zero-temperature exciton and trion linear absorption spectra for MoS$_2$, evaluated via

$$S(\omega) = \frac{2\pi}{\hbar}\sum_\alpha \left|\langle\Psi^i|V|\Psi^\alpha\rangle\right|^2 \Gamma(E^\alpha - E^i - \hbar\omega). \quad (2)$$

Here, $V = (eA/mc)\boldsymbol{\lambda}\cdot\mathbf{p}$ is the light–matter interaction, where $A$ is the vector potential, $e$ and $m$ are the electron charge and mass, $c$ is the speed of light, $\mathbf{p}$ is the momentum operator, and $\boldsymbol{\lambda}$ is the polarization of the optical field. Here we use circular polarization that selectively excites carriers in the $K$ valley. For the exciton spectrum, the sum is over all eigenstates of the electron–hole Bethe–Salpeter equation, whereas the initial state, labeled "i," is the Fermi vacuum. For the trion spectrum, the sum is over all eigenstates of the two-electron-one-hole Bethe–Salpeter equation, and the initial state has one excess electron at the minimum of the conduction band, as discussed above. Physically, this implies that the absorption event occurs such that an excess electron (i.e., due to negative doping) resides within the coherence length of the optically created electron–hole pair. For singlet (triplet) trions, this electron has spin down (up) and momentum $\mathbf{Q}$ at the $K'$ ($K$) point; see Fig. 1. Note that the optical selection rules involved in exciton and trion absorption are otherwise fully identical (in terms of momentum, spin, and valley degrees of freedom). The function $\Gamma$, containing the excited-state lifetime and other line-shape broadening effects, is taken to be a Lorentzian with a width of 4 meV. Shown in Fig. 2a are results for $N = 80$ and $k_0 = 0.10$, while spectra resulting from different convergence parameters are shown in Fig. 2b and in Supplementary Fig. 3.

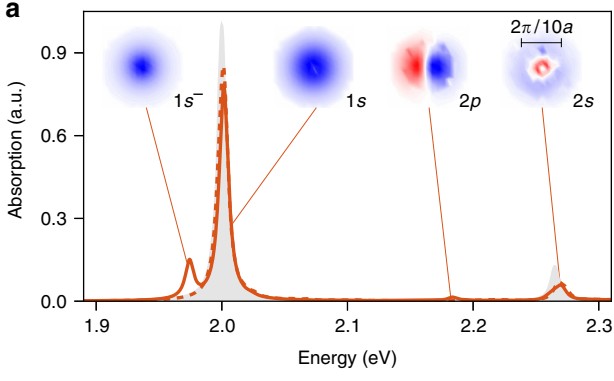

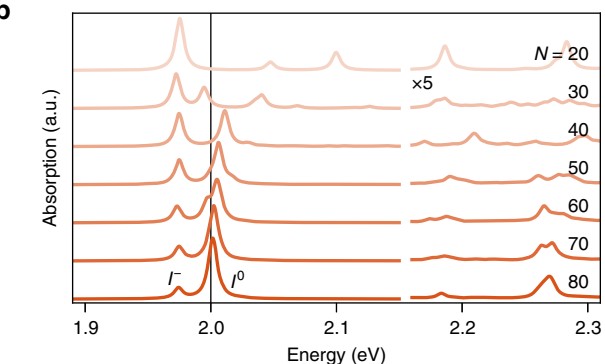

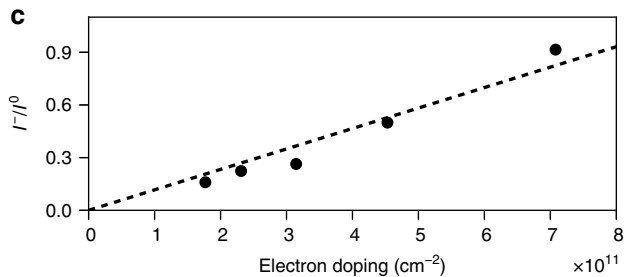

**Fig. 2** Exciton and trion absorption spectra and doping dependence.
**a** Calculated helicity-selective singlet (solid curve) and triplet (dashed curve) trion spectra of monolayer $MoS_2$. The two-particle exciton spectrum is shown in gray. For select singlet trion states, the wavefunction in reciprocal space near the $K$ point is shown as a heatmap. **b** Singlet trion spectra for varying Brillouin zone sampling resolutions, $N$. Vertical line at 2.0 eV depicts the converged exciton resonance energy. Trion and exciton resonance peak are labelled as $I^-$ and $I^0$, respectively. **c** Trion-to-exciton peak intensity ratio as a function of the electron doping density, calculated as $2/\sqrt{3}N^2a^2$ with the lattice constant given by $a = 3.193$ Å for $MoS_2$ (results are shown for $N \geq 40$). Linear fit with intercept at the origin shown as dash

The exciton spectrum in Fig. 2a shows typical behavior, with a pronounced 1s resonance at 2.00 eV and higher lying states corresponding to a non-hydrogenic Rydberg series[4]. Consistent with the discussion above, the singlet trion spectrum shows a peak located at ~1.97 eV corresponding to a bound trion, whereas the triplet trion spectrum shows no such peak. Importantly, in both trion spectra we find peaks coinciding with the neutral exciton energies. These can be understood as continuum resonances of the trion, corresponding to a free electron that is unbound from the exciton[48]. This picture is consistent with the k-space singlet trion wavefunctions that we also present in Fig. 2a, which are plotted as a function of the hole momentum with one of the electrons fixed at $K$ (the other electrons momentum is then equal to the hole momentum, due to momentum conservation). Besides showing an s-type azimuthal symmetry of the bound

singlet trion state, these wavefunctions confirm that the *entire* Rydberg series is reproduced by the trion calculations, including the broken degeneracy between 2s and 2p excitons[38] (the finite oscillator strength observed for the latter is the result of limited sampling, and disappears with increasing N). We conclude that the bound trion state and such exciton resonances share a common ground state, corresponding to a single excess electron at the conduction band minimum. This common ground state suggests that a coherent exciton–trion cross-peak should be observable in 2DES, which we turn to next, after briefly discussing the role of electron doping.

While the density of Brillouin zone sampling serves as a convergence parameter, it also provides a proxy to study the doping dependence. Our k-space calculations effectively involve a real-space periodic lattice with $N \times N$ unit cells over which a single excess electron is distributed in the initial state. Hence, with increasing N the electron doping density decreases. This is borne out in Fig. 2b where we show singlet trion absorption for varying N, which shows an intensity redistribution from trion-to-exciton with increasing sampling resolution. This behavior is quantified in terms of the effective doping density in Fig. 2c, showing a linear dependence of the trion-to-exciton peak intensity ratio to the electron doping density. We note that the simplified electronic structure used here (two bands and a static, model dielectric function) combined with the Brillouin zone truncation scheme enables us to study convergence behavior and low doping densities beyond that achievable by a fully first-principles approach and full Brillouin zone sampling[40].

**Exciton–trion coherence and spectral quantum beats**. We next turn our attention to 2DES, through which the coherent[17] and incoherent[16] dynamics of excited states can be monitored. In this technique, details of which can be found elsewhere[49,50], a material interacts with four ultrashort laser pulses, which can be grouped into an initial "excitation" pair and a subsequent "detection" pair, and the resulting signal is commonly presented as an excitation–detection correlation spectrum for each time delay between the two pulse pairs. Different combinations of pulse interactions result in different spectral signals. In our aim to interpret recent experiments on TMDCs[23], we specifically focus on the non-rephasing stimulated emission signal,

$$S(\omega_1, t_2, \omega_3) = -\left(\frac{2\pi}{\hbar}\right)^2 \sum_{\alpha,\beta} \left|\langle\Psi^i|V|\Psi^\alpha\rangle\right|^2 \left|\langle\Psi^i|V|\Psi^\beta\rangle\right|^2$$
$$\times e^{-(i\omega_{\alpha\beta} + \gamma_{\alpha\beta})t_2} \Gamma^*(E^\alpha - E^i - \hbar\omega_1)\Gamma(E^\beta - E^i - \hbar\omega_3).$$
$$(3)$$

Here, $\omega_1$ and $\omega_3$ are the excitation and detection energies, respectively, and $t_2$ is the time delay. The complex lineshape function is given by $\Gamma(\omega) = 1/(i\omega - \sigma)$, with $\sigma$ as the linebroadening parameter. The excitation energy difference between excited states $\alpha$ and $\beta$ is given by $\omega_{\alpha\beta} = (E_\alpha - E_\beta)/\hbar$ and $\gamma_{\alpha\beta}$ represents the associated decoherence rate. The most important mechanistic origin of decoherence is believed to be the scattering of quasiparticles with lattice phonons, and a microscopic investigation of this phenomenon is an interesting topic for future research, albeit beyond the scope of the present study. The damped oscillation as a function of the time delay (quantum beat) contained in the exponential is mapped onto the 2DES signal weighted by the product of transition matrix elements between the excited states and a common initial state $\Psi^i$. In particular, excited states that do not share a common initial state, as might arise in inhomogeneous samples, do not show coherent cross-peaks in 2DES.

Recently, Hao et al.[23] recorded time-dependent oscillatory signals in 2DES of electron-doped $MoSe_2$ monolayers at 20 K,

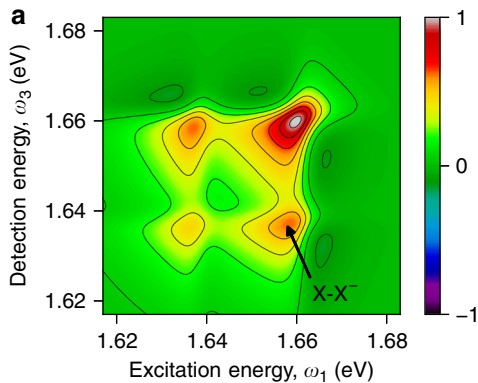

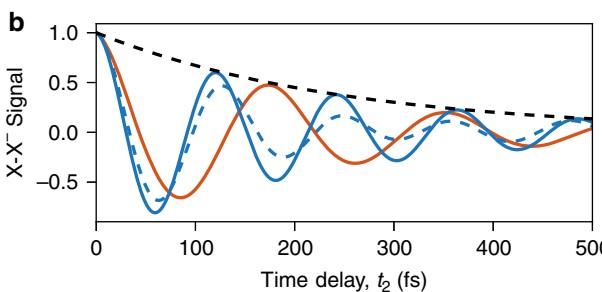

**Fig. 3** Exciton–trion quantum beats in two-dimensional spectroscopy. **a** Total (singlet plus triplet) trion two-dimensional electronic spectrum of monolayer $MoSe_2$, calculated through Eq. (3) at zero time delay ($t_2 = 0$ fs). **b** Time-dependent signal at the lower cross-peak (X–X⁻) location for $MoSe_2$ (red) and $WSe_2$ (blue), shown together with an exponential indicating the phenomenological exciton–trion coherence decay (black dashed). For $WSe_2$, results are shown for fully degenerate singlet and triplet trions (solid), and for which the triplet trion is blue-shifted by 7 meV as a result of exchange interactions (dashed). Shown in all cases are the real (absorptive) part of the complex signal

yielding indications of coherent interactions between the bound trion and the 1s exciton, and the observed quantum beat decay suggested an associated dephasing time of $\gamma_{X-X^-}^{-1} = 250$ fs. However, a reliable determination of the exciton–trion coherence time requires detailed knowledge of how such quantum beats are affected by possible interfering oscillatory signals. Our many-body formalism, in its ability to simulate 2DES, allows us to address this in a straightforward manner, while offering the prospect of microscopically investigating the decoherence mechanisms. For now we resort to a phenomenological treatment of the latter, and study the 2DES of TMDCs using $MoSe_2$ and $WSe_2$ as representative examples. In Fig. 3a, we show the sum of the singlet and triplet 2D spectrum for $MoSe_2$ resulting from Eq. (3) at zero time delay and with cocircularly polarized pulses, obtained for the same convergence parameters as in Fig. 2a. The impulsive signal is multiplied by a Gaussian laser spectrum centered at the bound trion state and with a standard deviation of 17 meV, accounting for the limited laser bandwidth affecting the experimental measurements[23]. The agreement with the 2DES measurements by Hao et al.[23] is excellent, with the spectrum showing four peaks in a square arrangement, resulting from two optical transitions readily identified with the bound trion and the 1s exciton.

According to Eq. (3), the quantum beats due to the exciton–trion coherence are mapped onto the cross-peaks corresponding to trion excitation and exciton detection, and vice versa. Indeed, these spectral locations were employed in the quantum beat measurements by Hao et al.[23]. However, as

discussed above, the quantum beats only result from pairs of states ($\alpha$ and $\beta$) that are optically coupled to a common initial state, $\Psi^i$. In Fig. 2a we observed the 1s exciton in all of the trion and exciton calculations, but only its resonance in the *singlet* trion configuration contributes to exciton–trion quantum beats observed for $MoX_2$, since only this resonance optically couples to the same initial state as the bound trion (an excess spin-down electron relaxed in the $K'$ valley). Hence, it is by virtue of the emergence of an exciton resonance in the trion (two-electron-one-hole) Hilbert space, that coherent interactions between excitons and trions arise.

Figure 3b shows the time-dependent signal of the lower (below-diagonal) cross-peak for $MoSe_2$ resulting from our model while imposing $\gamma_{X-X^-}^{-1} = 250$ fs (the other cross-peak, shown in Supplementary Fig. 4, in principle exhibits an identical signal except for incoherent contributions from population transfer that are not considered in our simulation). The signal features a pronounced oscillation, consistent with the measurements[23], with the oscillation period matching the Fourier inverse of the singlet trion binding energy. Consistent with the above discussion, we find this quantum beat to result from the bound singlet trion state coherently interacting with its exciton resonance. A comparison of the associated quantum beat decay with the reference decay function $e^{-t_2 \gamma_{X-X^-}}$ shows the destructive interference with auxiliary states to be negligible, such that the exciton–trion coherence time is indeed accurately reflected in this oscillatory signal. This substantiates that quantum dephasing in $MoSe_2$ induces a measurable coherence decay time of 250 fs, as inferred from the reported 2DES experiments[23].

For $WX_2$, both singlet and triplet trions form bound states, and as such both contribute to exciton–trion quantum beats resulting from coherent interactions with their respective exciton resonance. Previous reports have suggested that repulsive exchange interactions between conduction and valence band electrons breaks the degeneracy of the singlet and triplet trions by about 7 meV[44–47]. Inclusion of such interactions in the interaction kernel of the Bethe–Salpeter equation requires access to the real-space structure of the underlying orbitals, which our level of formalism does not provide. However, such interactions are known to be very local in real space and thus well approximated by a $k$-independent contribution to the Bethe–Salpeter equation solution[51]. In our simulation, we are flexible in including or excluding this contribution, thereby modulating the degeneracy of the singlet and triplet trions. If we exclude the interaction, the beating pattern shown in Fig. 3b is very similar to the $MoSe_2$ quantum beat, apart from a somewhat higher oscillation frequency (consistent with a higher binding energy). Again, the quantum beat decay is found to form a reliable probe of the underlying exciton–trion dephasing time. However, if we include an exchange interaction leading to an energy splitting of 7 meV, the beating signal changes appreciably, as shown in Fig. 3b. The nondegenerate trion states lead to an apparent destructive interference in the total (singlet plus triplet) quantum beat, as a result of which the beat decay occurs considerably faster than the actual dephasing time. Altogether, these results demonstrate that the quantum beats observed at the exciton–trion cross-peak locations in 2DES provide a lower bound to the actual exciton–trion coherence time, and that $WX_2$ in particular warrants caution because of the presence of two (nearly degenerate) trion species.

## Discussion

We have presented a many-body formalism for the simulation of time-resolved nonlinear spectroscopy including three-particle excitonic complexes. Although the formalism can be straightforwardly implemented in a first-principles manner, we have here

employed a parameterized two-band model and an isotropic, static dielectric function. Combined with a careful truncation of the Brillouin zone, these choices allowed us to provide highly converged results despite the otherwise high computational cost. In applying this formalism to excitons and trions, we uncover various fundamental properties of these charge carrier complexes that relate to the optoelectronic functionality of TMDCs and provide excellent agreement with recently measured 2DES. As noted before, helicity- and frequency-selective excitation of the $A$ transition in the $K$ valley allows control of the spin state of the optically created electron–hole pair. Consistent with our results, an even more comprehensive spin control can be achieved for bound trion states in $MoX_2$: helicity-selective excitation at the bound trion transition generates three-body complexes consisting of a spin-up hole, and spin-differing electrons (following the convention from Fig. 1). The resulting state coherently interacts with exciton resonances that optically couple to a shared ground state consisting of an excess spin-down electron relaxed in the $K'$ valley. In case of $WX_2$, where both singlet and triplet trions form bound states with near-degenerate transition energies, such selective excitation generates both well-defined spin configurations with ratios dictated by the spins of the doping charges, and each trion state coherently interacts with the exciton resonance with which it shares a one-electron ground state. In real space, such a sharing of a ground state can be thought of as the photoexcited electron–hole pair and the single electron residing within each others coherent domain. This is automatically fulfilled in theoretical models based on Bloch states, representative of pristine, extended crystals with translational symmetry, such as employed here. Nevertheless, actual materials are characterized by a certain degree of impurities and scattering with phonons, which break this symmetry and limits the size of coherent domains. The level of theory employed here in principle allows sufficient flexibility to include electron–phonon couplings parametrized against first-principles calculated electron–phonon coupling strengths[52]; for example, by means of a semiconductor Bloch equation approach[53] or Heine–Allen–Cardona approach[54,55], or perturbatively at the level of the Boltzmann transport equation. This would allow to unravel the microscopic origin of electronic decoherence and relaxation.

## Methods
**Many-body spectral calculations**. The theoretical methodology and parametrization employed in this work are described in detail in Supplementary Methods. Our many-body formalism can in principle be combined with an ab initio treatment of the full band structure of TMDCs. For the results presented here, we instead used the two-band model introduced by Xiao et al.[11], which through parametrizations against first-principles calculations allows to realistically account for the band structure in the $K$ and $K'$ valleys at arbitrary sampling resolution. The Brillouin zone was discretized using a Monkhorst–Pack grid, while a truncation radius was imposed around the $K$ and $K'$ points. Excitons and trions are described using the Bethe–Salpeter equation and its generalization to three-body systems[37]. The dielectric screening appearing in the Coulomb interaction term was taken to be of analytical form, $\varepsilon(q) = 1 + 2\pi\chi_{2D}q$, with the two-dimensional polarizability parametrized against first-principles calculations. Linear spectra were calculated using Fermi's Golden Rule, while 2D spectra were obtained through its extension to higher dimensions.

## Data availability
Figures 2 and 3 and S1–S4 have associated raw data. All relevant data is available by written request to the authors.

## Code availability
The code used for simulations is available by written request to the authors.

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

## Acknowledgements

This work was supported in part by the Air Force Office of Scientific Research under award number FA9550-18-1-0058. The Flatiron Institute is a division of the Simons Foundation. We thank Alexey Chernikov and David Reichman for helpful discussions.

## Author contributions

R.T. and T.C.B. conceived the project. R.T. developed the simulation code and performed the calculations. R.T. and T.C.B. interpreted the results and wrote the paper.

## Additional information

**Competing interests:** The authors declare no competing interests.

