## [Peer Review File · Nature Communications]

Reviewers' comments:

Reviewer #1 (Remarks to the Author):

This manuscript describes a new computational approach for predicting time-dependent nonlinear spectroscopy. The method involves the construction of matrix elements between states that are 2- and 3-particle solutions to the Bethe-Salpeter equation (and extensions of the BSE). Using a two-band model and a model dielectric function, the authors apply their approach to MoS₂ and WS₂, computing excitons and trions accurately, and then generating 2d spectra in good agreement with experiment from which exciton-trion coherences are predicted and elucidated, and new selection rules are uncovered for the evolution of trions from photoexcited singlets.

The manuscript is well written, and the framework and results are clearly explained. The transition metal dichalcogenides are important systems of broad interest, and this work explains complex and rather subtle effects in their time-resolved nonlinear spectroscopy. I recommend publication, after the below two points are addressed.

The results in Fig. 1 appear to summarize calculations with a 2 band model. However, they are not clearly presented as the results of calculations performed in this work. If Fig. 1 summarizes prior work, that work should be cited; if it summarizes computations new to this work, that should be indicated. It should also be explained somewhere why two bands are sufficient for this system; and why electron-hole interactions, which usually mix vertical transitions over the Brillouin zone, simplify in this case to the cartoon in Fig. 1.

The authors should discuss more completely the scenario and selection rules for which a triplet is photo excited, such as those that feature nontrivial absorption in Fig. 2. This is somewhat unconventional.

The phenomenological damping parameter in Eq. 3 is not discussed in a thorough manner. Could the authors explain the physics of this parameter? Should a single parameter suffice in theory, even if it works in practice?

Reviewer #2 (Remarks to the Author):

The manuscript gives a (partial) theoretical description of the four wave mixing experiment in Ref.[23] on MoSe₂, which is the main motivation for the work. This is based on a Fermi Golden rule description of Four Wave mixing via eq.(3) in the submitted manuscript.

To this end, the trions and excitons wave-functions (WFs) and energies (E) are plugged inside eq.(3). WFs and E are computed within a many-body approach applied to a two band model, also using a model screened interaction. Instead the dephasing-times entering eq.(3) are just a parameter.

The calculation of trions in TMDs is an interesting topic. They are difficult to compute in an ab-initio manner and the use of a model system is a good alternative. In the results it is also shown how trions enter absorption of doped samples.

However, beside the calculation of trions WFs and E, the manuscript is a very simple analysis of the results given by equation 3 for MoS₂ and WS₂ which could be obtained directly solving it with simple parameters for E and WFs integrals.

Also the resulting description of Ref.[23] is only "partial" for a number of reasons:

- TMDs different from the one measured in Ref.[23] are considered
- the approach presented here completely misses the different signal of the opposite cross peaks measured in the experiment
- the de-phasing time is not computed
- energies and WFs are obtained by model calculations (thus they cannot be fully quantitative) and some effects like exchange interaction are added only "a posteriori"

There are thus no significant insights obtained by the accurate solution of the two bands model and, in my opinion, the results are not worth for consideration in this journal.

Reviewer #3 (Remarks to the Author):

This manuscript presents calculations of the optical absorption spectra and the two-dimensional electronic spectroscopy (2DES) of MoS₂ and WS₂. The exciton and trion states are calculated from the solution of the Bethe-Salpeter equation for both quasiparticles (as in Ref. [34] for trions). The electronic structure is obtained with a parametrised 2 band model and the screening Coulomb interaction is modelled following Ref. [38]. For MoS₂, the results from the simulations lead to similar conclusions as from the experiment on MoSe₂ in Ref [23]. On the other hand, for WS₂ the simulations predict a “destructive interference emerging from coexisting singlet and triplet trions” and that the quantum beat does not unambiguously reflect the exciton-trion coherence time but provides a lower bound for it.

To my understanding, the original part of this work is the calculation of the 2DES through Eq. 3. The major claim is the fact that due to the “destructive interference emerging from coexisting singlet and triplet trions”, the quantum beat does not unambiguously reflect the exciton-trion coherence time (Figure 3b, solid and dash green curves).

In general, the theory, methodology and calculations are technically sound. The manuscript is clearly written and accessible to a broad audience. The length is appropriate. I think that the results are of interest to others in the field (spectroscopy of and physics of quasiparticles in two-dimensional materials) and may have implications for the employment of these materials for optoelectronics. Technical details of the methodology and calculations may be of interest to theoreticians working on the simulations of optical spectroscopies.

More specifically, I would like the authors to clarify several points before considering whether to recommend the manuscript for publication in Nature Communications.

First, regarding the claim of “destructive interference emerging from coexisting singlet and triplet trions”, it is not entirely clear to me how the “intravalley exchange interaction” was included in the model and how this is justified. For example, why can one choose a k-independent intravalley exchange interaction? Considering this effect is key to the main claim of the work, I would suggest that the authors include more details and a discussion in the Supplementary Material (SM).

At this regard, the exchange interaction was neglected in the interaction kernel of the Bethe-Salpeter equations. In the SM, I could not find a discussion on why this effect is neglected. In fact, it seems to me that it is the exchange causes the splitting that is key to the central claim of the manuscript. In a first principles framework, the cost of calculating the exchange part is not higher than that of the Coulomb screening part of the quasiparticle interaction kernel, so from my experience, I cannot see the rationale for not including this effect in the model.

Second, considering that the experiment was performed for MoSe₂, why do they authors not include this system in their study?

Third, there are few claims in the manuscript that are not completely accurate and I would like the authors to review them in an eventual revision of the manuscript.

(a) In the abstract, the authors claim to “present a many-body formalism for the simulation of time-resolved nonlinear spectroscopy”

I think that this is a slightly on the overselling side as the manuscript focusses on one particular type of spectroscopy and is specialised to exciton/trions interaction. Further, the manuscript does not address a generalization of the method presented and does not seem to me that the development of methodology is central to the manuscript. I would then suggest to reformulate this claim so to reflect better the achievements and aims of the manuscript.

(b) In the introduction, the authors claim that “there is a lack of first-principles techniques capable of simulating nonlinear optical response of condensed matter materials”.

Though, I do agree that the state-of-the-art for nonlinear optical response is not at the level of that of linear spectroscopies, I think that this statement does not fairly reflect the progresses made in the last 5-10 years.

For example, real-time approaches to time-dependent density-functional theory (TD-DFT) have been recently extended to treat periodic crystals and used for e.g. simulate high-harmonic generation in solids (see e.g. Nature Communications vol 8, 745 (2017)). Within a response TDDFT framework, nonlinear properties such as the second-harmonic generation or the electro-optical effect have been calculated for semiconductors even including excitonic effect from a simple kernel interaction model (see e.g. Phys. Rev. B 97, 205201 (2018), Phys. Rev. B 82, 235201 (2010)). On the many-body side, the first extensions to the nonlinear response I am aware of are Phys. Rev. B 57, 6519 (1998) and Phys. Rev. B 65, 035205 (2002). More recently, the Bethe-Salpeter equation has been used to study 2D materials in Phys. Rev. B 89, 235410 (2014) and a real-time approach Bethe-Salpeter equation was developed (Phys. Rev. B. 84 245110 (2011)) and applied to nonlinear optical spectroscopies (e.g. for 2D materials Phys. Rev. B 98, 165126 (2018), Phys. Rev. B 89, 081102(2014)) and time-resolved spectroscopy (e.g. Nano Lett. 17 (8):4549-4555 (2017), Phys. Rev. B 93, 195205 (2016)).

I recognize that all the examples above regarding neutral excitations which cannot be used directly for the present work. I certainly do not ask the authors to add such detailed discussion on first-principles nonlinear optical response and all those references (which are a few picks) and I even wonder whether mentioning nonlinear optical response, in general, is relevant for introducing their work. I think they should either review their statement to better reflect the state-of-the-art or discuss more specifically the case of charged excitations and spectroscopies relevant to the work.

(c) In conclusion, the authors claim that the inclusion of electron-phonon coupling in their model is straightforward. Considering the importance of this effect (see e.g. Ref [23]), one wonders then why the effect was not included. Otherwise, the authors may want to add a few details that help the reader to see how this effect can be included straightforwardly. Do they have in mind something similar to what is proposed in Phys. Rev. Lett. 101, 106405? Or else?

In addition, there are minor points I would like the authors to consider:

- p.5. eq. (2): A is described as the vector potential but is a scalar. Is the same A appearing in the fraction in front of the interaction kernel in Eq. S7?
- In the SM, Fig S1, not clear which curves refer to the trion and which to the exciton.
- In the SM, the authors should add references for (S10)
- In the SM, p. S7, how large are the shifts the authors applied to the calculated spectra in Fig. 2?

Reviewers' comments:

Reviewer #1 (Remarks to the Author):

This manuscript describes a new computational approach for predicting time-dependent nonlinear spectroscopy. The method involves the construction of matrix elements between states that are 2- and 3-particle solutions to the Bethe-Salpeter equation (and extensions of the BSE). Using a two-band model and a model dielectric function, the authors apply their approach to MoS2 and WS2, computing excitons and trions accurately, and then generating 2d spectra in good agreement with experiment from which exciton-trion coherences are predicted and elucidated, and new selection rules are uncovered for the evolution of trions from photoexcited singlets.

The manuscript is well written, and the framework and results are clearly explained. The transition metal dichalcogenides are important systems of broad interest, and this work explains complex and rather subtle effects in their time-resolved nonlinear spectroscopy. I recommend publication, after the below two points are addressed.

We thank the reviewer for expressing his/her appreciation of the submitted work, and for raising a number of insightful remarks. We have addressed these remarks as specified below.

The results in Fig. 1 appear to summarize calculations with a 2 band model. However, they are not clearly presented as the results of calculations performed in this work. If Fig. 1 summarizes prior work, that work should be cited; if it summarizes computations new to this work, that should be indicated. It should also be explained somewhere why two bands are sufficient for this system; and why electron-hole interactions, which usually mix vertical transitions over the Brillouin zone, simplify in this case to the cartoon in Fig. 1.

Indeed, Fig. 1 is a cartoon of the low-energy band structure, which serves as a common language for Dirac-like materials including TMDCs with spin-orbit coupling. We have adjusted the caption of Fig. 1, clarifying that it serves as a cartoon illustration of the spin dependent band structure near the K points derived from previously reported first-principles calculations (Ref. 12 in the revised manuscript), and that vertical transitions become mixed as a result of quasiparticle interactions. We have also motivated on page 4 the application of the two-band model based on negligible differences compared to more sophisticated methodologies.

The authors should discuss more completely the scenario and selection rules for which a triplet is photo excited, such as those that feature nontrivial absorption in Fig. 2. This is somewhat unconventional.

First we would like to emphasize that the "triplet" moniker refers to the spin configuration of the two electrons in the three-particle trion complex and not to the spin multiplicity of the overall state. With spin-orbit coupling, S^2 is not a good quantum number. This is not a singlet-to-triplet transition, which is indeed typically forbidden. All bright excitations conserve S_z as expected. We have added a more thorough discussion of the scenario and selection rules involved in triplet absorption to pages 5-6 of the revised manuscript. In principle, the selection rules underlying the photo-excitation of a trion state are identical to that of an exciton in terms of the spin, momentum, and helicity degrees of freedom. The sole difference is that whereas for excitons absorption results from exciting onto the vacuum state, for trions an initial one-electron state is present within the coherence length of the optically created electron-hole pair, and the singlet or triplet label of the trion depends on whether the two electrons have opposite or equal spin, respectively. While the singlet trion features a bound three-body state below the lowest-energy exciton state, such three-body state is unstable for triplet trions in molybdenum-based compounds owing to exchange interactions. This explains why in Fig. 2 we find a red-shifted trion peak for the singlet configuration, whereas in the triplet configuration no such peak is to be found (with the resulting spectrum being identical to the excitonic counterpart).

The phenomenological damping parameter in Eq. 3 is not discussed in a thorough manner. Could the authors explain the physics of this parameter? Should a single parameter suffice in theory, even if it works in practice?

The damping parameter models the effect of electron-phonon interactions, which produce coherence decay through population relaxation and pure dephasing. Unfortunately, the ab initio inclusion of electron-phonon interactions is a present-day challenge; almost all many-body calculations of *linear* absorption spectra (e.g. using the GW/BSE framework) employ a phenomenological linewidth, similar to our own. We believe our formalism presented here can be extended to include perturbative electron-phonon interactions (at the level of the Boltzmann transport equation), though this advance is beyond the scope of the present work. We have added a brief discussion of this to the revised manuscript on pages 6-7 and 10.

Reviewer #2 (Remarks to the Author):

The manuscript gives a (partial) theoretical description of the four wave mixing experiment in Ref.[23] on MoSe₂, which is the main motivation for the work. This is based on a Fermi Golden rule description of Four Wave mixing via eq.(3) in the submitted manuscript.

To this end, the trions and excitons wave-functions (WFs) and energies (E) are plugged inside eq.(3). WFs and E are computed within a many-body approach applied to a two band model, also using a model screened interaction. Instead the dephasing-times entering eq.(3) are just a parameter.

The calculation of trions in TMDs is an interesting topic. They are difficult to compute in an ab-initio manner and the use of a model system is a good alternative. In the results it is also shown how trions enter absorption of doped samples.

However, beside the calculation of trions WFs and E, the manuscript is a very simple analysis of the results given by equation 3 for MoS₂ and WS₂ which could be obtained directly solving it with simple parameters for E and WFs integrals.

Also the resulting description of Ref.[23] is only "partial" for a number of reasons:

- TMDs different from the one measured in Ref.[23] are considered
- the approach presented here completely misses the different signal of the opposite cross peaks measured in the experiment
- the de-phasing time is not computed
- energies and WFs are obtained by model calculations (thus they cannot be fully quantitative) and some effects like exchange interaction are added only "a posteriori"

There are thus no significant insights obtained by the accurate solution of the two bands model and, in my opinion, the results are not worth for consideration in this journal.

We thank the reviewer his/her comments, and for underscoring the importance of studying trion states in TMDCs. We do however disagree with the reviewer's view that no significant insights are obtained as well as with his/her characterization of the computational methodology. For *linear* absorption spectroscopy of semiconductors, the GW/BSE framework is currently the state of the art and has produced innumerable insights into the excitonic properties of 2D materials (see, for example, Nat. Mater. 13, 1091 (2014); Nature 513, 214 (2014); Phys. Rev. Lett. 121, 167402 (2018)). However, this formalism is "simply" calculating exciton wavefunctions and energies and plugging them into the lowest-order Fermi's Golden Rule expression for absorption. To our knowledge, our manuscript presents one of the first proposals and implementations to use similarly-obtained wavefunctions and energies in the nonlinear (four-wave mixing) Golden Rule

expressions, and to furthermore employ these to analyze a class of recent state-of-the-art experiments.

A concern raised by the reviewer is that our description of the experimental measurements by Hao et al. (formerly Ref. 23) is only partial and we appreciate his/her points. In response, we have significantly expanded upon the originally reported numerical results.

- We have repeated our full set of calculations for selenium-based TMDCs as studied in Hao et al.'s experiments. The new results are fully in line with the qualitative conclusions drawn in the original submission, but now form an integral part of the revised manuscript in order to keep quantitative differences compared to experimental conditions minimal.
- We have included the suggested analysis of the signal at the opposite cross-peak location in the Supplementary Information as a newly added figure S5, again keeping our analysis as close as possible to the experimental conditions.
- An ab initio calculation of the dephasing time is interesting, but beyond the scope of the present manuscript, which is aimed at understanding and interpreting the manifestation of trion states in linear and nonlinear spectroscopy, rather than the microscopic dephasing mechanisms that dictate their quantum dephasing (see also our response to Reviewer #1 above).
- With regard to our “model” calculation and the quantitative accuracy of our results: we emphasize that the adopted choices are purely made for convenience and physical insight. In particular, the formalism presented can be performed in a fully ab initio way, but with a greater computational expense. Furthermore, the parameters of our model (band gap, carrier effective mass, and dielectric properties) are extracted from ab initio simulation, and thus there are no “free” parameters, except for the exchange interaction considered near the conclusion. The use of such a band-structure parameterization within a GW/BSE framework has been shown to yield quantitative agreement with more expensive fully ab initio calculations (Ref. 11 in the revised manuscript). As an interesting technical point, our work demonstrates that an extremely dense sampling of the Brillouin zone is required for convergence and Brillouin zone sampling at this density is currently beyond the reach of fully ab initio approaches.

We again thank the reviewer for his/her comments, which have allowed us to significantly improve the submitted manuscript by redoing our calculations for selenium-based TMDCs and adding an extended analysis of the cross-peak oscillations. We also hope to have clarified the significance of the submitted work.

Reviewer #3 (Remarks to the Author):

This manuscript presents calculations of the optical absorption spectra and the two-dimensional electronic spectroscopy (2DES) of MoS₂ and WS₂. The exciton and trion states are calculated from the solution of the Bethe-Salpeter equation for both quasiparticles (as in Ref. [34] for trions). The electronic structure is obtained with a parametrised 2 band model and the screening Coulomb interaction is modelled following Ref. [38]. For MoS₂, the results from the simulations lead to similar conclusions as from the experiment on MoSe₂ in Ref [23]. On the other hand, for WS₂ the simulations predict a “destructive interference emerging from coexisting singlet and triplet trions” and that the quantum beat does not unambiguously reflect the exciton-trion coherence time but provides a lower bound for it.

To my understanding, the original part of this work is the calculation of the 2DES through Eq. 3. The major claim is the fact that due to the “destructive interference emerging from coexisting singlet and triplet trions”, the quantum beat does not unambiguously reflect the exciton-trion coherence time (Figure 3b, solid and dash green curves).

In general, the theory, methodology and calculations are technically sound. The manuscript is clearly written and accessible to a broad audience. The length is appropriate. I think that the results are of interest to others in the field (spectroscopy of

and physics of quasiparticles in two-dimensional materials) and may have implications for the employment of these materials for optoelectronics. Technical details of the methodology and calculations may be of interest to theoreticians working on the simulations of optical spectroscopies.

More specifically, I would like the authors to clarify several points before considering whether to recommend the manuscript for publication in Nature Communications.

We thank the reviewer for expressing his/her appreciation of the broad appeal, technical level, and scholarly presentation of the manuscript, and for his/her thorough and thoughtful comments that have been of great help to further strengthen the manuscript and Supplementary Information. In the following, we detail our responses.

First, regarding the claim of “destructive interference emerging from coexisting singlet and triplet trions”, it is not entirely clear to me how the “intravalley exchange interaction” was included in the model and how this is justified. For example, why can one choose a k-independent intravalley exchange interaction? Considering this effect is key to the main claim of the work, I would suggest that the authors include more details and a discussion in the Supplementary Material (SM).

At this regard, the exchange interaction was neglected in the interaction kernel of the Bethe-Salpeter equations. In the SM, I could not find a discussion on why this effect is neglected. In fact, it seems to me that it is the exchange causes the splitting that is key to the central claim of the manuscript. In a first principles framework, the cost of calculating the exchange part is not higher than that of the Coulomb screening part of the quasiparticle interaction kernel, so from my experience, I cannot see the rationale for not including this effect in the model.

The exchange interaction is known to be very local in real-space and thus reasonably constant in k-space; see for example “Calculation of the Exchange Energy for Excitons in the Two-Body Model”, Rohner, Phys. Rev. B (1971). Therefore, we believe the approximation of constant exchange in k-space is reasonable. Regarding the neglect of exchange in the BSE: the referee is correct that in a first-principles context, the cost of exchange is no worse than RPA screening of the direct interaction. However, our mean-field problem is essentially a tight-binding model with parameterized interactions to reproduce the many-body (e.g. GW) band structure and thus we have no access to the real-space structure of the underlying orbitals. Whereas the direct interaction can be calculated in a coarse-grained manner without the underlying orbital structure, the exchange interaction is zero in the same approximation, and requires an empirical parameter. We have added a discussion of this to the revised manuscript on pages 8-9.

Second, considering that the experiment was performed for MoSe₂, why do they authors not include this system in their study?

We have redone all reported calculations for selenium-based compounds, results of which now form an integral part of the revised manuscript, including the presented 2D spectral data.

Third, there are few claims in the manuscript that are not completely accurate and I would like the authors to review them in an eventual revision of the manuscript.

(a) In the abstract, the authors claim to “present a many-body formalism for the simulation of time-resolved nonlinear spectroscopy”

I think that this is a slightly on the overselling side as the manuscript focusses on one particular type of spectroscopy and is specialised to exciton/trions interaction. Further, the manuscript does not address a generalization of the method presented and does not seem to me that the development of methodology is central to the manuscript. I would then suggest to reformulate this claim so to reflect better the achievements and aims of the manuscript.

We have rewritten the abstract in the revised manuscript, in order to optimally convey the manuscript's core message.

(b) In the introduction, the authors claim that “there is a lack of first-principles techniques capable of simulating nonlinear optical response of condensed matter materials”.

Though, I do agree that the state-of-the-art for nonlinear optical response is not at the level of that of linear spectroscopies, I think that this statement does not fairly reflect the progresses made in the last 5-10 years.

For example, real-time approaches to time-dependent density-functional theory (TD-DFT) have been recently extended to treat periodic crystals and used for e.g. simulate high-harmonic generation in solids (see e.g. Nature Communications vol 8, 745 (2017)).

Within a response TDDFT framework, nonlinear properties such as the second-harmonic generation or the electro-optical effect have been calculated for semiconductors even including excitonic effect from a simple kernel interaction model (see e.g. Phys. Rev. B 97, 205201 (2018), Phys. Rev. B 82, 235201 (2010)). On the many-body side, the first extensions to the nonlinear response I am aware of are Phys. Rev. B 57, 6519 (1998) and Phys. Rev. B 65, 035205 (2002). More recently, the Bethe-Salpeter equation has been used to study 2D materials in Phys. Rev. B 89, 235410 (2014) and a real-time approach Bethe-Salpeter equation was developed (Phys. Rev. B. 84 245110 (2011)) and applied to nonlinear optical spectroscopies (e.g. for 2D materials Phys. Rev. B 98, 165126 (2018), Phys. Rev. B 89, 081102(2014)) and time-resolved spectroscopy (e.g. Nano Lett. 17 (8):4549-4555 (2017), Phys. Rev. B 93, 195205 (2016)).

I recognize that all the examples above regarding neutral excitations which cannot be used directly for the present work. I certainly do not ask the authors to add such detailed discussion on first-principles nonlinear optical response and all those references (which are a few picks) and I even wonder whether mentioning nonlinear optical response, in general, is relevant for introducing their work. I think they should either review their statement to better reflect the state-of-the-art or discuss more specifically the case of charged excitations and spectroscopies relevant to the work.

We thank the reviewer for bringing these works to our attention. We further agree that a thorough discussion of these developments is beyond the scope of the present work, but that a reformulation is warranted. We note that although various optical techniques probe the third-order susceptibility, 2DES uniquely involves the complete set of third-order dynamical pathways, providing access to an unmatched level of physical information. To our knowledge, our manuscript is the first many-body presentation of the fully coherent four-wave mixing signals underlying 2DES. We have modified the sentence in question in the revised manuscript on page 2, and added citations to a number of the references suggested.

(c) In conclusion, the authors claim that the inclusion of electron-phonon coupling in their model is straightforward. Considering the importance of this effect (see e.g. Ref [23]), one wonders then why the effect was not included. Otherwise, the authors may want to add a few details that help the reader to see how this effect can be included straightforwardly.

Do they have in mind something similar to what is proposed in Phys. Rev. Lett. 101, 106405? Or else?

We consider the inclusion of electron-phonon coupling an important topic of future research, albeit beyond the scope of the present study. We agree that providing a few details regarding such implementation is a valuable addition to the manuscript, and have added a mentioning of these to the Conclusion section of the revised manuscript.

In addition, there are minor points I would like the authors to consider:

- p.5. eq. (2): A is described as the vector potential but is a scalar. Is the same A appearing in the fraction in front of the interaction kernel in Eq. S7?

The factor A appearing in Eq. S7 is not the vector potential, but instead is the area of the 2D Brillouin zone. We have clarified this in the revised SM on page S4.

- In the SM, Fig S1, not clear which curves refer to the trion and which to the exciton.

In the revised SM, we have added arrows to figure S1 and the newly introduced figure S2 to resolve this.

- In the SM, the authors should add references for (S10)

We have added the referencing to the revised SM.

-In the SM, p. S7, how large are the shifts the authors applied to the calculated spectra in Fig. 2?

Generally, these shifts depend on the applied Brillouin zone truncation parameters, but are typically in the range 0.50 - 0.75 eV. We have described this on page S7 of the revised SM.

Reviewers' comments:

Reviewer #1 (Remarks to the Author):

I think authors have more than answered my questions and those of the other reviewers. I recommend publication.

Reviewer #2 (Remarks to the Author):

In my previous review I pointed out one main issue:

MI- beside the calculation of trions WFs and E, the manuscript is a very simple analysis of the results given by equation 3 for MoS₂ and WS₂ which could be obtained directly solving it with simple parameters for E and WFs integrals

And a list of other issues:

I1- TMDs different from the one measured in Ref.[23] are considered

I2- the approach presented here completely misses the different signal of the opposite cross peaks measured in the experiment

I3- the de-phasing time is not computed

I4- energies and WFs are obtained by model calculations (thus they cannot be fully quantitative) and some effects like exchange interaction are added only "a posteriori"

Reply

I1: The authors have improved the manuscript addressing I1.

I2: It was due to a wrong comparison I did of the experimental plot of ref[32] against the theoretical plot.

I3: It has not been addressed. I agree the computation is beyond what can be done abinitio. However, since the goal here is to give a description of an experimental results, it remains a limitation of the manuscript

I4: I'm satisfied by the authors reply. Let me say that the approach is quasi-quantitative.

MI: Concerning the main issue. Since the authors explained that the results are quasi-quantitative, this also partially addresses it. The main result is then that their quasi-quantitative approach confirms finite matrix elements between the initial state and both the trion and the exciton. Accordingly the existence of cross signals is confirmed by the theory, also giving the correct energy splitting between the trion and the exciton.

However, this rises a new concern. What is the initial state entering equation (3) chosen? Since the Trion is an excitation of a state with N+1 particles while the Exciton of a state with N particles, it cannot be a simple state. Is it a linear combination of the ground state and a state with one extra electrons? If so, how are the coefficients of such linear combination chosen. I would expect such coefficient to be related to the doping of the sample and accordingly the ration between the "X-X" peak and the "X-T" peak.

Even if I3 remains not addressed, the authors should clarify this last point, i.e. how good is the semi-quantitative estimation of the (relative) intensity of the "X-T" peak (compared to the "X-X" peak) as a function of the doping. I'd very much appreciate to see 1d plots of the peaks intensity, i.e. vertical cuts of the 2D spectrum at $w=X$. After such clarification, the manuscript can be accepted in my opinion.

Reviewer #3 (Remarks to the Author):

The authors replied satisfactorily to my comments and modified accordingly the manuscript and SM adding more calculations, clarifying a few technical points and amending a few statements that were not fully accurate.

In view of those changes and of the observations in my previous review regarding the manuscript, I recommend the manuscript for publications in NCOMMS.

Reviewer #2 (Remarks to the Author):

In my previous review I pointed out one main issue:

MI- beside the calculation of trions WFs and E, the manuscript is a very simple analysis of the results given by equation 3 for MoS2 and WS2 which could be obtained directly solving it with simple parameters for E and WFs integrals

And a list of other issues:

I1- TMDs different from the one measured in Ref.[23] are considered

I2- the approach presented here completely misses the different signal of the opposite cross peaks measured in the experiment

I3- the de-phasing time is not computed

I4- energies and WFs are obtained by model calculations (thus they cannot be fully quantitative) and some effects like exchange interaction are added only "a posteriori"

Reply

I1: The authors have improved the manuscript addressing I1.

I2: It was due to a wrong comparison I did of the experimental plot of ref[32] against the theoretical plot.

I3: It has not been addressed. I agree the computation is beyond what can be done abinitio. However, since the goal here is to give a description of an experimental results, it remains a limitation of the manuscript

I4: I'm satisfied by the authors reply. Let me say that the approach is quasi-quantitative.

MI: Concerning the main issue. Since the authors explained that the results are quasi-quantitative, this also partially addresses it. The main result is then that their quasi-quantitative approach confirms finite matrix elements between the initial state and both the trion and the exciton. Accordingly the existence of cross signals is confirmed by the theory, also giving the correct energy splitting between the trion and the exciton.

However, this rises a new concern. What is the initial state entering equation (3) chosen? Since the Trion is an excitation of a state with N+1 particles while the Exciton of a state with N particles, it cannot be a simple state. Is it a linear combination of the ground state and a state with one extra electrons? If so, how are the coefficients of such linear combination chosen. I would expect such coefficient to be related to the doping of the sample and accordingly the ration between the "X-X" peank and the "X-T" peak.

Even if I3 remains not addressed, the authors should clarify this last point, i.e. how good is the semi-quantitative estimation of the (relative) intensity of the "X-T" peak (compared to the "X-X" peak) as a function of the doping. I'd very much appreciate to see 1d plots of the peaks intensity, i.e. vertical cuts of the 2D spectrum at $w=X$. After such clarification, the manuscript can be accepted in my opinion.

We thank the referee once again for his/her thorough inspection of the submitted work. We are pleased that our revisions have adequately resolved the issues raised in the previous report. The referee has raised an additional concern related to the nature of the initial state from which excitons and trions become optically excited.

It is true that trions live in a N+1 particle Hilbert space, and that an N particle Hilbert space solely yields excitons. However, and importantly, exciton states are also observed in the N+1 particle Hilbert space. Such states are excitons unbound to an excess electron that are stable

excitations living in the two-electron-one-hole subspace, that have been discussed in earlier works such as Ref. 48. Hence, the exciton-trion cross-peaks (and the underlying coherent interactions) emerge from the trion and “exciton” resonance in the $N+1$ particle Hilbert space, coupling through the same initial state, namely the vacuum state plus one excess electron. We have expanded on this important point in the revised manuscript on pages 5 and 7.

This brings us to an important point that was not explicitly made in the previous submissions, namely that of the role of doping. Having one excess electron while varying the sampling resolution (denoted N in our manuscript - not to be confused with the number of particles) effectively means varying the electron doping density. In real space, the sampling resolution determines the number of TMDC unit cells in real-space as N^2 . Increasing N therefore corresponds to decreasing the doping density, which in turn impacts the spectral peak intensities. The impact on 2D spectroscopy is not easily quantified because of laser pulse spectrum effects, but linear spectroscopy provides a fairly unambiguous way to demonstrate this behavior. We have therefore added a new panel to what was formerly Figure S3, and merged the resulting figure with Figure 2 in the main text, showing the trion to exciton peak ratio from linear absorption as a function of doping density (derived from N^2) for MoS₂. This confirms that the relative trion peak increases monotonously with enhanced doping, depending linearly on the electron doping density (although small deviations are found to arise from limited sampling of the exciton and trion states). Hence, increasing N yields both more accurate excited states and reduced doping densities.

Besides adding the aforementioned figure panels to Figure 2, we have inserted a discussion of the doping dependence in the main text on page 7. We have also clarified that the coherent exciton-trion interactions arise from the emergence of exciton resonances in the trion Hilbert space in the revised manuscript on page 9.

REVIEWERS' COMMENTS:

Reviewer #2 (Remarks to the Author):

The authors have now addressed all major point.
The manuscript can now be accepted in my opinion.